# The Influence of Male Biostimulation on Cloacal Anatomy and Egg-Laying Behavior in Young Female Muscovy Ducks (*Cairina moschata forma domestica*)

**DOI:** 10.3390/ani14132002

**Published:** 2024-07-07

**Authors:** Martin Linde, Axel Wehrend, Abbas Farshad

**Affiliations:** 1Veterinary Clinics for Reproductive Medicine and Neonatology, Justus-Liebig-University of Giessen, 35392 Giessen, Germany; tierarzpraxis-linde@gmx.de (M.L.); axel.wehrend@vetmed.uni-giessen.de (A.W.); 2Laboratory of Reproduction Biology, Department of Animal Science, Faculty of Agriculture, University of Kurdistan, Sanandaj 6617715175, Iran

**Keywords:** biostimulation, cloaca, ducks, egg-laying, Muscovy, reproduction

## Abstract

**Simple Summary:**

This study analyzed the effects of male biostimulation on female Muscovy ducks before egg-laying. The results demonstrated that the biostimulated female ducks were influenced before egg-laying and laid eggs sooner than the isolated ducks. Moreover, it was demonstrated that changes in cloacal morphology can be utilized as a differentiating characteristic among ducks in either the pre- or post-laying phase. In summary, this study establishes a basis for improving the productivity of Muscovy duck farms.

**Abstract:**

The importance of Muscovy ducks in industrial poultry production is growing; however, little is known about the physiology of their reproductive cycles. This study investigated the influence of male biostimulation on female ducks before the commencement of the laying phase. A total of 30 muscovy ducks, hatched in the same year at 289–341 days of age, were divided into two groups of 15 birds each and kept with and without contact with a male duck until the day of first egg-laying—319 ± 14 and 335 ± 13, respectively. Before reaching egg-laying maturity, the cloacae of 29 adult ducks were subjected to daily clinical assessments. The evaluations yielded four unique categories of outcomes, determined by assessing factors such as the degree of redness and protrusion of the mucous membrane, the moisture level, and swelling of the cloacal sphincter muscle. The results of this study on biostimulation revealed that, on average, female ducks that had contact with males laid their first egg 16 days earlier, weighing 78.7 ± 3.0 g, compared to the isolated female ducks, weighing 79.1 ± 7.0 g. Furthermore, there was no significant difference observed in the mean initial egg weight between the groups (*p* = 0.841). The cloacal morphology indicated significant morphological changes 25–26 days before laying. Efforts to improve Muscovy production and develop biotechnological techniques to modify these ducks’ reproductive cycle will benefit from these advancements.

## 1. Introduction

Biostimulation refers to the utilization of natural, advantageous, and cost-effective techniques involving the use of olfactory, visual, and auditory cues for interaction within the same species, which were first reported in ungulates in 1969 [1,2,3,4] and can impact neuroendocrine and behavioral responses, ultimately affecting reproductive efficiency and ovulation synchronization in various animal species [5]. According to research, interactions among individuals in nature can have a significant effect on reproductive physiology [6], which is shown in domestic and farm animals, laboratory rodents, and primates, to synchronize their reproductive activities with others of the same species [7].

Originally defined as the influence of a male on a female specimens’ sexual status, biostimulation has become an intricate field. Recent findings indicate that biostimulatory effects extend beyond the impact of males on females [2,8,9]; females also elicit a reproductive response to female–male and female–female interactions, where female chemical signals play important roles in sexual attraction [10,11,12,13,14,15]. According to a literature review, research on sheep, pigs, and goats demonstrates a noteworthy impact of biostimulation [16,17,18,19]. Although the male–female effect in cattle interactions has been demonstrated to affect female reproductive activity, it has received little attention [20]. Consequently, this practice has not yet become the standard for farms [9]. El-Azzazi et al. [21] noted a slight improvement in reproductive performance due to the male effect.

In addition, numerous studies have been conducted on the female reproductive cycles of various poultry species, including domestic poultry, leading to a comprehensive understanding of the physiological and anatomical alterations in the female genital tract throughout the sexual maturation and reproductive process in these species [22,23,24,25]. In this context, the reproductive activity and effectiveness of chickens is significantly influenced by various environmental factors, including stress, housing conditions, the availability of feed and water, and daily and seasonal changes [26,27,28]. Interestingly, studies investigating conspecific biostimulation in waterfowl are lacking. However, a previous study demonstrated that male turkeys play a role in the quantity of parthenogenetically fertilized eggs [29]. Additionally, in terms of king penguins, it has been reported that ambient noise coming from both their colony and, to a lesser degree, a different colony can affect sexual behavior and ovulation control, ultimately determining when eggs are laid. These environmental factors ultimately influence the reproductive success of king penguins with respect to egg-laying [30].

There is limited information available on the reproductive physiology of Muscovy ducks, and no studies have been conducted on modifying their reproductive cycle, despite the various significant roles played by other poultry species in global production [31]. The Muscovy duck (*Cairina moschata forma domestica*) is also playing an increasingly important role in industrial poultry production. This study, therefore, aimed to assess whether the reproductive performance of female Muscovy ducks can be improved by biostimulation through exposure to male conspecifics. Additionally, investigations were conducted to determine variations in cloacal conditions before the beginning of the laying period and create a clinical examination method to forecast the points of laying.

## 2. Materials and Methods

All the data were collected on a breeding farm as part of routine procedures and, therefore, do not require animal welfare approval under German law. This research involved the use of 35 Muscovy ducks, consisting of 5 males and 30 females, which were chosen from a lineage with the goal of minimizing genetic variation among the individuals. Each duck was artificially hatched in uniform breeding conditions and reared separately according to their sex. At the beginning of this study, the female ducks had not exhibited any egg-laying behavior. The male ducks were identified using unique footrings from the Federal Association of German Poultry Breeders (BDRG) and color-coded plastic rings. Female ducks were also marked with BDRG foot rings, and individually numbered plastic wing tags were attached to their right-wing membranes. The ducks were housed in sex-segregated groups before observation.

### 2.1. Testing Mature Conspecific Males for Biostimulation

Prior to the start of this experiment, all the animals were separated into groups based on their genders. The investigation commenced by involving two fully grown drakes, aged approximately 24 and 36 months, and three juvenile ducks that had been hatched alongside the female ducks, in relation to the utilization of male ducks which had been previously used. A total of 30 female ducks, hatched within a 60-day, were randomly allocated to six outdoor aviaries, each measuring 6 × 1.5 m. Each aviary was equipped with a one-meter-long feeding trough, a ten-liter inverted drinker, and a nesting box of specific dimensions. Subsequently, fifteen ducks were divided into three aviaries, each with five birds, arranged so that, from the beginning of the study until the laying period, a sexually mature male duck could always be housed in a neighboring enclosure. During this time, the ducks had contact with the male ducks and constantly interacted tactilely, visually, auditorily, and olfactorily, while another group of fifteen ducks was divided into three more aviaries, with five animals each. These aviaries were placed at a distance of about 100 m, without any possibility of contact with male ducks. Consequently, all ducks were subjected to identical climatic and light conditions. In addition, the length of days it took each duck to lay its first egg was accurately documented, along with the weight of the egg, to determine whether the inclusion of a male duck had a positive effect on the number of eggs produced and the weight of the first egg. Finally, the five male ducks were housed separately in 6 × 1.5 m enclosures, adjacent to three of the duck enclosures (Figure 1).

### 2.2. Determination of Cloacal Morphology

The cloacal morphology of the experimental ducks was determined by a systematic external examination of the cloaca on alternate days as soon as the first duck had indicated sexual maturity by laying their first egg (Figure 2). A duck exhibited a persistent incidence of cloacal mucosa, leading to its exclusion from the data collection. In this procedure, the animal being examined was secured upside down between the legs of the seated examiner. The tail was bent over the duck’s back using the middle, ring, and little fingers of the left hand, while the cloaca was exposed from the plumage with the thumb and index finger of the same hand. The parameters to be evaluated included the level of redness and edema of the musculus sphincter cloacae (MSC), the degree of MSC opening during the tail bending over the duck’s back, the level of moisture and redness of the cloacal mucosa, and the extent to which the cloacal mucosa bulged out when the tail was bent through the cloacal opening. The data were systematically recorded, sorted, and appraised in readiness for the start of egg-laying (Table 1). Subsequently, the findings were categorized based on the subjective classification system, specific animal, and day of examination. After the initiation of the egg-laying activity, the collected data were converted into days before the commencement of laying to analyze the morphological changes preceding this activity.

### 2.3. Detection of Laying Activity

To detect the act of laying eggs, the nesting boxes were examined during a time frame from 8 to 9 a.m. to determine if any eggs were present. If an egg was found, each animal within the associated aviary underwent digital screening to identify any egg-laying activity. The examiner then used their left hand to secure the wings of the animal at the level of the humeri and gently pressed them onto the ground. Subsequently, the examiner cautiously inserted the middle, ring, or little fingers from the right hand into the cloaca. When a duck had laid an egg, a still shell-free and plump-elastic egg could be felt through the cloacal wall on the right side of the belly.

### 2.4. Statistical Analysis

The data collected during this study were meticulously recorded in Excel^®^ data sheets, while the statistical inquiries were computed utilizing the SPSS 15.0 statistical program (SPSS Software GmbH, Munich, Germany). In order to evaluate the influence of the biostimulatory effects and the hatching date, a *t*-test was performed to compare the disparities in the onset of laying and the initial weight, aiming to identify any significant differences (*p* < 0.05) in the recorded parameters. Differences were considered significant if *p* < 0.05.

## 3. Results

### 3.1. Biostimulation by Male Counterparts

The research results demonstrated that all 30 sexually mature female Muscovy ducks began laying eggs, and each duck showed the presence of a second egg, without a shell, in the left dorsocaudal abdominal region. At 289–341 days of age, 15 ducks housed in visual, olfactory, and auditory contact with a male counterpart laid their first egg. The eggs were laid by the isolated ducks on days 312–358 of their lives. The average age of the eggs laid by the drake-stimulated ducks was 16 days earlier than the solitary females (319 ± 14 vs. 335 ± 13 days of age, respectively, *p* = 0.003). The first-laid eggs in the drake-contact group weighed an average of 78.7 ± 3.0 g. Nonetheless, the average weight of the eggs without drake contact was 79.1 ± 7.0 g, which did not differ significantly (*p* = 0.841) from the eggs with drake contact (Table 2).

### 3.2. Cloacal Anatomy before Laying

The results demonstrated in Table 3 indicate that significant alterations in the cloaca’s morphology were visible as early as on the 25th or 26th day before the start of laying. Every female duck showed a class 1 morphology. After the 25th and 26th day before laying, 75% of the ducks were still classified as class 1. The proportion of animals with class 3 cloacae significantly increased prior to the first or second day before egg-laying (*p* = 0.003). Up until the 9th or 10th day before the first egg, the number of class 1 cloacae drastically decreased (*p* < 0.001). Most animals (97%) demonstrated signs of a class 3 cloaca two-to-three days before their first egg-laying. Prior to egg-laying, only 13% of the ducks showed a class 4 cloaca, whereas, following the laying process, 97% demonstrated a class 4 cloaca. The increase in class 4 cloacae on the third day before egg-laying was deemed statistically significant (*p* < 0.001).

## 4. Discussion

Physiological and structural alterations in the female genital tract during sexual development and the reproductive cycle in domestic poultry species are well-documented [24,25]. Previous studies have demonstrated the advantageous effects of conspecific biostimulation on puberty onset, oestrus stimulation, synchronization, and ovulation rate in various species, such as sheep [17], goats [19], pigs [18,32,33], and cattle [34,35,36]. This biostimulation technique is a convenient and cost-effective means of enhancing the reproductive performance of domestic mammals. Interestingly, the effects of male conspecific biostimulation in domestic poultry have not yet been investigated. However, it has been found that acoustic signals can influence the egg-laying behavior of penguins [30]. A popular and economical method for improving female reproductive performance in domestic mammals is the use of male counterparts for biostimulation [16]. In particular, a combination of auditory, visual, and olfactory stimuli elicited the strongest response in sheep [37] and pigs [33]. 

The primary objective of this study was to investigate the effect of male biostimulatory effects on the initiation of egg-laying in young female Muscovy ducks. This study additionally assessed the cloacal conditions before egg-laying to develop predictive clinical examination techniques. The current investigation demonstrated that female ducks exposed to male ducks through biostimulation experienced a mean advancement of 16 days in laying their initial egg compared to female ducks that had been isolated. Moreover, the isolated ducks without male interaction exhibited significant changes in their cloacal morphology 25–26 days prior to egg-laying. This suggests that Muscovy ducks are sensitive to environmental influences such as male effects through biostimulation. Conversely, research indicates that variables such as feed and water availability, housing conditions, stress, and seasonal and daily variations significantly influence reproductive activity and efficiency [38,39], supporting the results of the current investigation. Research has shown that the presence of males can speed up the egg-laying process in females. It is essential to note that the reproductive cycle of domestic poultry is controlled by hormones. In birds, unlike mammals, the selection of a follicle is not influenced by neighboring follicles’ inhibitory or stimulatory effects. The maturation and differentiation of follicles in the ovary are regulated by the follicle-stimulating hormone (FSH) and the vasoactive intestinal peptide (VIP) from the gastrointestinal tract. Moreover, the FSH and the VIP play a role in facilitating cellular processes within the follicles through paracrine and autocrine mechanisms [40]. Therefore, this description suggests that hormonal changes due to the presence of males may act as a biostimulation factor in accelerating the reproductive cycle of Muscovy ducks.

The efficient diagnosis of approaching or occurring oestrus is of great importance for farm animals among domestic mammals, especially in terms of efficient insemination or controlled mating. The clinical examination of the external and internal female genitalia plays a significant role in this matter [41]. However, in poultry, an examination to predict or diagnose ovulation and the onset of the laying period has not yet been described. A crucial aspect of this study consisted, therefore, in determining whether it was possible to predict the onset of laying through a clinical examination of the female genitalia of Muscovy ducks. The anatomical structures to be examined were defined, and their relevant characteristics were divided into four diagnostic classes. This study provides evidence of the biostimulatory effect of male conspecifics on waterfowl. However, it remains unclear whether specific auditory, visual, or olfactory cues act independently or in conjunction to accelerate the onset of puberty. Previous studies in mammals have demonstrated that female reproductive performance is enhanced by olfactory, tactile, visual, and auditory cues from male conspecifics [16,34]. In addition, an ongoing investigation revealed that it is possible to predict the initiation of egg-laying in Muscovy ducks by conducting a clinical assessment of the cloaca. Similar to previous studies conducted on cattle and pigs, in which the examination of external genitalia was used to determine the optimal timing for artificial insemination [42,43], it was feasible to select ducks with grade 4 cloaca for either artificial insemination or natural mating from a larger population of breeding animals. This approach eliminates the need to maintain small breeding groups comprising only one male and a maximum of ten females. A significant number of ducks can, therefore, be paired with a duck for line breeding without sacrificing one or two eggs for incubation, considering the fact that insemination is performed only after the first egg is laid. In contrast, ducks that prioritize fertilized eggs from the beginning no longer require multiple-blind inseminations; instead, they may be selectively inseminated shortly before the initiation of anticipated egg-laying. The cloacal examination additionally enabled the selection of a large number of mother ducks into smaller groups. Considering that animals at different reproductive stages have varying dietary energy, nutrient, and mineral requirements [44], ducks may be fed more accurately based on their specific needs. Furthermore, ducks with grade 1 cloacae, which are still at least 10 days away from egg-laying, may be selected and stimulated towards earlier maturity through biostimulatory effects such as light programs or exposure to males. Ultimately, in order to assess the impact of male Muscovy ducks on the reproductive success of females, visual, auditory, and olfactory stimuli were employed. Should the beneficial outcomes of biostimulation in Muscovy ducks be validated, it is imperative to explore selective stimulation methods in future research endeavors [45].

## 5. Conclusions

To sum up, the improvements in Muscovy duck breeding, along with the utilization of biotechnological techniques to regulate the reproductive cycle, offer substantial advantages. The findings indicate that young Muscovy ducks, when exposed to a male drake, commenced egg-laying approximately 16 days earlier on average, whereas their initial egg weight remained unaffected. This early onset of egg-laying has important implications for the cost-effectiveness of large breeding flocks. By starting egg production earlier, these ducks have the opportunity to increase their income and reduce production costs. In addition, the earlier start of egg-laying allows for a faster turnover of eggs, resulting in more frequent sales and potentially higher profits. Furthermore, the shorter time to egg production helps reduce the overall costs associated with rearing ducks until they reach laying age. Therefore, optimizing the time to start laying can have a positive impact on both the financial profitability and efficiency of duck farming. However, this study did not examine whether certain stimulus variables led to an early onset of puberty, and more research is needed to address this issue. Consequently, it is plausible that the collective influence of these stimuli contributes to some of the effects observed in ducks. Further research is, therefore, necessary to ascertain the extent to which tactile cues stimulate Muscovy ducks and the relative significance of each stimulus in females.

## Figures and Tables

**Figure 1 animals-14-02002-f001:**
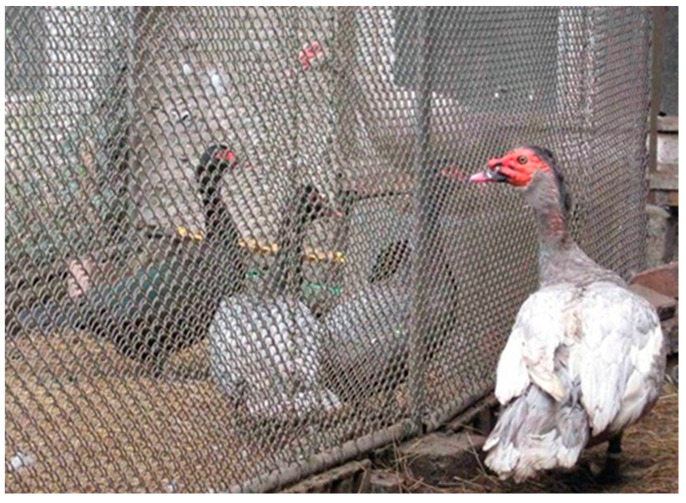
Demonstration of ducks (left) housed with contact with a drake (right) enabled by an olfactory, visual, and acoustic connection through a wire mesh fence.

**Figure 2 animals-14-02002-f002:**
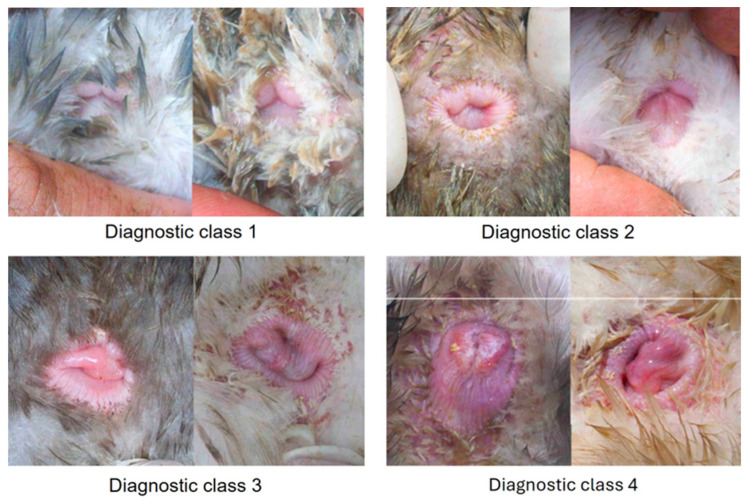
Display of various diagnostic cloacae classes, including low-grade (1), medium-grade (2), high-grade (3), and musculus sphincter (4).

**Table 1 animals-14-02002-t001:** Description of subjective diagnostic classes 1–4 as low-grade (lg), medium-grade (mg), high-grade (hg), and musculus sphincter cloacae (MSC), respectively.

	Class 1	Class 2	Class 3	Class 4
Redness of MSC	none	lg	mg	mg–hg
Edema of MSC	none	lg	mg	mg–hg
Opening of MSC	none	lg	mg	hg
Moisture in cloacal opening	none	no	lg	mg–hg
Redness of cloacal mucosa	none	lg	mg	mg–hg
Protrusion of cloacal mucosa	none	no	lg	mg–hg

**Table 2 animals-14-02002-t002:** Mean and standard deviation of the age of female Muscovy ducks at the onset of egg-laying activity in days with or without drake contact.

Nr.	Age at Egg-Laying (Days)	Weight of First Egg (Grams)	Age at Egg-Laying (Days)	Weight of First Egg (Grams)
1	289	81	312	70
2	299	78	321	79
3	309	81	323	76
4	310	82	325	79
5	313	73	328	80
6	315	75	329	79
7	316	79	333	73
8	322	78	333	98
9	322	82	333	79
10	325	83	338	78
11	325	78	342	78
12	327	77	347	87
13	333	75	350	82
14	337	82	350	81
15	341	77	358	68
	319 ± 14	78.7 ± 3.0	335 ± 13	79.1 ± 7.0 g

There was a statistically significant difference between the two groups, each comprising 15 ducks (*p* = 0.003). However, the average egg weight was not significantly different between the two groups, with 15 ducks each (*p* = 0.841) (*n =* 30).

**Table 3 animals-14-02002-t003:** Cloacal findings in female Muscovy ducks during the 58 days before the onset of egg-laying (*n =* 29).

Days before the Onset of Laying	Class 1 *n*/%	Class 2 *n*/%	Class 3 *n*/%	Class 4 *n*/%
58–27	197/100	0/0	0/0	0/0
26–25	19/ 75	6/25	0/0	0/0
24–23	21/84	4/16	0/0	0/0
22–21	14/70	6/30	0/0	0/0
20–19	13/59	9/41	0/0	0/0
18–17	9/36	16/64	0/0	0/0
16–15	9/32	19/68	0/0	0/0
14–13	3/12	21/81	2/8	0/0
12–11	1/4	14/58	9/38	0/0
10–9	0/0	14/50	14/50	0/0
8–7	0/0	9/33	18/67	0/0
6–5	0/0	4/14	24/86	0/0
4–3	0/0	1/3	28/97	0/0
2–1	0/0	2/6	26/81	4/13
Laying	0/0	0/0	1/3	28/97

The classification of diagnostic classes 1 to 4 includes low-grade (lg), medium-grade (mg), high-grade (hg), and musculus sphincter cloacae (MSC), respectively.

## Data Availability

The data included in this study are the property of the authors and can be obtained by contacting the corresponding author at afarshad@uok.ac.ir or abbas.farshad@vetmed.uni-giessen.de upon reasonable request.

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
