# Peer review of "The Influence of Male Biostimulation on Cloacal Anatomy and Egg-Laying Behavior in Young Female Muscovy Ducks (Cairina moschata forma domestica)"

_animals, 2024, doi:10.3390/ani14132002_

Round 1

Reviewer 1 Report

Comments and Suggestions for Authors

Comments to Authors

The influence of male biostimulation and clinical methods on egg laying in young female Muscovy ducks (Cairina moschata forma domestica)

LN: Line Number

LN2: Please delete: ....and clinical methods....

LN3: Please add: Cloacal anatomy and...

LN3: Please add: behaviour

LN2-3: Propose to revise the title as:  The influence of male biostimulation on cloacal anatomy and egg laying behaviour in young female Muscovy ducks (Cairina moschata forma domestica)

LN21: Please mention the age, BW & SD of the females used in the study.

LN22: Please mention the age and BW of the bird.

LN23: Why 29: Please explain.

Ln32: Propose to arrange alphabetically.

LN97-99: Please mention the BW ± SD of each 15-group dicks. (As the cloacal anatomy and egg size/production can be influenced by the size/BW of the birds)

LN163-164: Table 2: Please enter the value for Age at egg laying± SD for those who held without male. The value has been presented at the next column (Rt.)

LN163: Table 2: Please indicate the correct value for Weight of first egg ± SD: for the ducks held without male

Table 2: Please mention the sample number used in footnotes.

LN167: Please revise as Cloacal anatomy before laying

LN193: Table 3: Suggest to use: onset instead of ''start''

LN252: Not in the reference list.

LN260-262: This statement should be the one which appeared as the first statement of conclusion.

LN304: Reference no 40 is not mentioned in the text. Reference no: 43 is not in the list of references. Please check and revise.

Comments on the Quality of English Language

Good. Need minor editing. 

Author Response

We sincerely thank you for reviewing this manuscript and for the constructive comments and suggestions. Please find the detailed responses below and the corresponding revisions/corrections highlighted in Yellow /in track changes in the re-submitted files. The manuscript has been extensively revised to align with the review's findings, with meticulous highlighting of the corrections in yellow. Moreover, we have diligently attended to the raised suggestions and queries.

1: LN2-3: Propose to revise the title as: The influence of male biostimulation on cloacal anatomy and egg laying behavior in young female Muscovy ducks (Cairina moschata forma domestic)

Answer: The title has been edited and rewritten as suggested.

 2: LN21: Please mention the age, BW & SD of the females used in the study.

Answer: Done!

 3: LN22: Please mention the age and BW of the bird.

Answer: Done in the manuscript!

 4: LN23: Why 29: Please explain.

Answer: A duck exhibited a persistent incident of cloacal mucosa, leading to its exclusion from the data collection. Consequently, the cloacae of 29 mature ducks were routinely examined through daily clinical assessments.

5: Ln32: Propose to arrange alphabetically

Answer: Keywords have been edited and arranged alphabetically.

6: LN97-99: Please mention the BW ± SD of each 15-group dicks.  (As the cloacal anatomy and egg size/production can be influenced by the size/BW of the birds)

Answer: Done and highlight in the manuscript.

 7: LN163-164: Table 2: Please enter the value for Age at egg laying±SD for those who held without male. The value has been presented at the next column (Rt)

Answer: The mistakes and needed additions in table 2 have been corrected and edited.

 8: LN163: Table 2: Please indicate the correct value for Weight of first egg ± SD: for the ducks held without male

Answer: The mistakes in table 2 have been edited and corrected.

 9: Table 2: Please mention the sample number used in footnotes

Answer: The sample numbers have been edited and added.

10: LN167: Please revise as Cloacal anatomy before laying

Answer: The paragraph’s title has been edited and corrected as suggested.

11: LN193: Table 3: Suggest to use: onset instead of ''start''

Answer: Edited and replaced.

12. LN252: Not in the reference list.

Answer: Corrected and highlighted.

12: LN260: In the conclusion section, there is a comment highlighted without suggestion

Answer: This sentence has been edited and corrected.

13: LN304: Reference no 40 is not mentioned in the text. Reference no: 43 is not in the list of references. Please check and revise.

Answer: The text has been revised to correct references 40 and 43, both in the body of the text and in the reference list.

Reviewer 2 Report

Comments and Suggestions for Authors

My general opinion about the work is negative. I recommend that the article be rejected.

Author Response

We sincerely thank you very much for reviewing this manuscript and for the constructive comments and suggestions. We also regret that we were not able to convince reviewer 2 of our results. We believe that our study has produced important results for this breed that are not found in the existing literature. Despite this, we have addressed some of the criticisms, please find the detailed responses below and the corresponding revisions/corrections highlighted in green /in track changes in the re-submitted files.

Title: The influence of male biostimulation and clinical methods on 2 egg laying in young female Muscovy ducks (Cairina moschata 3 forma domestica)

In this study, the effects of male biostimulation before egg laying on female Muscovy ducks were determined.

Simple Summary: It is written in accordance with the content of the study and is understandable.

Abstract: It is written in accordance with the content of the study and is understandable.

Keywords: It is written in accordance with the content of the study and is understandable.

Introduction:

It should be further emphasized why the study was done on ducks. What kind of beneficial or harmful effects will it have on Muscovy ducks? It would be beneficial to add brief explanations.

Answer: We endeavored to modify the paragraph by elaborating on our approach to classifying the ducks to improve efficiency. It is important to mention that the rationale for choosing this breed is located in the introduction. Previous studies on waterfowl exist. We selected Muscovy ducks because they are a readily available species and hold economic significance.

How did we determine the number of 5 male ducks? The number was found to be very low. It would make sense if 5 men were used per treatment, at least individually.

Answer: It is essential to highlight that we did not carry out any statistical precalculations concerning the necessary number of ducks in the context of this research. Due to the absence of preliminary studies, there was a lack of data in the literature to compute a larger sample size. Furthermore, the remaining reviewers did not criticize the number of animals used.

Unfortunately, the material and method are not technically understandable. Having difficulty understanding.Answer: We endeavored to thoroughly revise the recommendation in a comprehensive manner. The specific recommendations for improvement made by the other reviewers have been implemented, and it is our hope that this section has become more comprehensible as a result.

Which analysis was applied to which data should be written. Statistical analysis is lacking in this aspect

Answer: We have revised and updated the statistical analysis section in response to this request and suggestion.

Reviewer 3 Report

Comments and Suggestions for Authors

Overall this read like a small-scale student project and was written by a group of students. Would like to see more in-depth discussion in introduction and discussion section. Recommend several changes - please see attached.

Comments on the Quality of English Language

Some spelling mistakes.

Author Response

We express our sincere gratitude for reviewing this manuscript and the constructive comments and suggestions.  The manuscript has been extensively revised to align with the review's findings, with meticulous highlighting of the corrections in purple. Moreover, we have diligently attended to the raised suggestions and queries.

1: 14-15 The reader will not know which birds are in Groups 2, 3, and 4, or what these Groups mean upon first reading the simple summary. Recommend to define groups first or remove names of groups.

Answer: As suggested, the section edited.

2. 74-77 Hypothesis missing?

Answer: As suggested, done and highlighted.

3. 86 How old were the 3o juvenile ducks?

Answer: we tried to explain to make it understandable

4. 96-104 From the sentence “subjected to the presence of a mature male”, it is not clear whether the male was kept in the same aviary, or adjacent. Furthermore, the article states 15 ducks in one pen, and 15 in another, but a total of 33 female ducks were reported. Lastly. The article also states 5 drakes were used. But here, only 1 drake was used. What was the purpose of the extra (unused) ducks and drakes?

Answer: We attempted to compose, amend, revise, and emphasize in the text as recommended by you and other reviewers the portion to enhance clarity for the readers.

5. 119-121 Table 1. Research articles should contain detailed information whereby it can be easily reproduced upon reading the materials and methods. However, it is not clear how each of these classes listed in Table 1 was classified. What constitutes redness, edema, opening, moisture, etc? How can a scientist objectively measure this? Was a colourimeter / colour scale used? What about moisture? What counts as low-, medium-, and high-grade?

Answer: To ensure improved clarity for the readers, we endeavored to compose, amend, revise, and emphasize the text based on your suggestions and the feedback provided by other reviewers.

6. 133-137 What is your experimental design? What is your experimental unit? At what level was significance declared?

Answer: We attempted to revise and describe the introduction and section 2.1 in the material method to make the content more understandable for the readers of the manuscript reviewers.

7. 140-141 It is better to state the age that the ducks started laying, since “March and May” will not be clear to the reader.

Answer: We tried to amend and clarify in the Manuscript section 3.1 to make the content more understandable.

8. 141-142 “Second egg … which was laid in the left dorsocaudal abdominal region” sentence awkward, especially since ‘laid’ means oviposition. Do you mean to say the presence of the second egg can be felt in the left dorsocaudal abdominal region?

Answer: Edition and correction have been done!

9. 146-147 Reporting of P-values can be more concise, i.e. “The average age of eggs laid by drake-stimulated ducks was 16 days earlier than solitary females (319 vs 335 days of age, respectively, P=0.003). It is not a major change, so I will leave it to the author to decide if this change is warranted.

Answer: The valuable suggestion has been utilized.

10. 164 Average + SEM missing for ducks age at egg-laying in ducks held without male? Additionally, in the last row first column (under cell “15”), insert “Average ± SEM”. What is the thin line between Numbers 5 and 6? Please double check that the table follows the Animals guidelines.

Answer: The correction has been done!

11. 165 “tow” misspelling

Answer: The correction has been done!

12. 168 What is “external clinical examination”? Do you just mean assessment of the cloaca?

Answer: Yes, we mean the assessment of the cloaca subjectively!

13. 171 What makes a cloaca “clinically significant”? I am having trouble understanding the terms, and why they were used. Recommend to change or clarify.

Answer: Edition and correction have been done!

14. 173 “desribed” misspelling

Answer: The edition has been done!

15. 168-190 Overall, this paragraph was quite hard to read and follow, since the timeline was presented as counting down to oviposition, and not chronologically. Furthermore, this data including Table 3 presents ducks from both groups as a single treatment, but it should be split. The way in which the data was analysed leaves the reader confused. Was this section analyzed statistically? Or is the author just reporting scores 2 months prior to egg laying? Comparing the “clinical diagnosis” between groups at different ages would have been more beneficial to answer this research’s objective.

Answer: To enhance readability and overall quality, this paragraph has been rewritten!

16. 194 Table 3 – line spacing in the headers need to be fixed.

Answer: To enhance readability and overall quality, this paragraph has been rewritten!

17. 196-255 The discussion section read very much like a methods paper, indicating that the clinical assessments worked well to determine how many more days until egg laying.

However, discussion between the two main treatments (exposure to males vs not) is

recommended, since that was the main objective of the paper. Would be great to include more underlying physiological mechanisms / hormone / pheromones that result from exposure of males to females. Explain the results – why did exposure to males speed up oviposition in females?

Answer: we have tried to edit to improve the discussion section.

  1. 245 “sire” -> “drake”?

Answer: In Replaced as suggested.

Reviewer 4 Report

Comments and Suggestions for Authors

Manuscript animals-3065550, entitled “The influence of male biostimulation and clinical methods on egg laying in young female Muscovy ducks (Cairina moschata forma domestica)

Recommendation:       The above paper is not suitable for publication in its present form.

The article provides useful information about the effects of male biostimulation on egg laying in young female Muscovy ducks.

My main concern is that the experimental design is not clear. How many females were housed in each aviary? Which was the ratio female:male? Was the same male used in each group of females within the whole experimental period? When was the beginning of your study? At what age of the ducks?

How many were the ducks? 29 as in L23, 36 as in L82 or 30 as in L140? Why were three ducklings aged 10 weeks used?

Some parts are repeated in introduction and discussion. Please minimize repetitions

What is the meaning of “and clinical methods” in the title?

L14-15: Please explain what do you mean by group 2, 3 and 4.

L21-23: The sum of 15+15 is not 29.

L28: Please delete “both”

L36: “interaction” instead of “communication”

L37: “…which were firstly observed in…”

L38: “impact” instead of “affectimpact”

L41: What do you mean by “reproductive information is shared among”?

L63-65: How is this study related with male biostimulation?

L69-72: The parts of this sentence are not connected

L80-81: I think that every study including treatment with animals needs animal welfare approval

L107: Please specify the days

Please check the title of Table 1 (L119-120) and correct.

L144-145: What do you mean? Please clarify

Please correct the last row of Table 2. The number “335 ± 13” should be added in the previous column and the weight of first egg (g) should be added

L174: What do you mean by “After the 26th and 25th day before egg laying”?

L182: 2 to 3 or 4 to 3?

In Table 3, please define classes as a footnote

L258: “muscovite”?

L259: Did you develop “biotechnological techniques”?

Figure 2 is not clear and it is not necessary, since these results are already presented in Table 3. Please delete

Comments on the Quality of English Language

Moderate editing of English language required

Author Response

We express our sincere gratitude for reviewing this manuscript and the constructive comments and suggestions offered by reviewer 4. The manuscript has been extensively revised to align with the review's findings, with meticulous highlighting of the corrections in turquoise. Moreover, we have diligently attended to the raised suggestions and queries.

1: My main concern is that the experimental design is not clear. How many females were housed in each aviary? Which was the ratio female:male? Was the same male used in each group of females within the whole experimental period? When was the beginning of your study? At what age of the ducks?

Answer: The questions pertaining to the utilization of ducks, individual aviaries, female-to-male ratios, and other relevant aspects have been reviewed and enhanced within the materials and methods section

2: How many were the ducks? 29 as in L23, 36 as in L82 or 30 as in L140? Why were three ducklings aged 10 weeks used?

Answer: We endeavored to enhance the effectiveness of the study by providing a detailed account of our methodology for categorizing the ducks, thereby attempting to edit and revise the section pertaining to the number of ducks

  1. Some parts are repeated in introduction and discussion. Please minimize repetitions

Answer:  We made an effort to edit and rewrite the discussion as closely as possible, following the suggestions provided.

4. What is the meaning of “and clinical methods” in the title?

Answer: Edited and corrected.

5. L14-15: Please explain what do you mean by group 2, 3 and 4:

In section “Determination of cloacal morphology” explained.

L21-23: The sum of 15+15 is not 29:

Edited and corrected.

L28: Please delete “both”

Edited and corrected. L36: “interaction” instead of “communication”:

Edited and Corrected.

L37: “…which were firstly observed in…”:

Edited and Replaced with “reported”.

 L38: “impact” instead of “affectimpact”:

Edited and Replaced.

 L41: What do you mean by “reproductive information is shared among”?:

It refers to the communication between various genders in a species.

L63-65: How is this study related with male biostimulation?

Although the study is unrelated to male biostimulation, it is referenced here as an illustration of biostimulation in birds.

L69-72: The parts of this sentence are not connected;

Edited and corrected.

L80-81: I think that every study including treatment with animals needs animal welfare approval:

The manuscript has been updated to include the Institutional Review Board Statement and Informed Consent Statement at the end

6. Which analysis was applied to which data should be written. Statistical analysis is lacking in this aspect

Answer: We have revised and updated the statistical analysis section suggestion.

7. L107: Please specify the days

Answer: Edited.

8. Please check the title of Table 1 (L119-120) and correct.

Answer: Edited and corrected.

9. L144-145: What do you mean? Please clarify

Answer: Edited and corrected.

10. Please correct the last row of Table 2. The number “335 ± 13” should be added in the previous column and the weight of first egg (g) should be added

Answer: Edited and corrected.

 11. L174: What do you mean by “After the 26th and 25th day before egg laying”?

Answer: They indicated the days of pre-laying eggs. The values have been edited and corrected.

12. L182: 2 to 3 or 4 to 3?

Answer: The values 2 to 4 or 4 are the identification class of cloaca morphological classes. The paragraph has been edited to be understandable.

13. In Table 3, please define classes as a footnote

Answer: Edited and corrected.

14. L258: “muscovite”?

Answer: Edited and corrected.

15. L259: Did you develop “biotechnological techniques”?

Answer: Concerning this question, we think that our results present a methodological advancement, illustrating that this biostimulation technique has a positive effect on early egg laying in Muscovy ducks. This outcome is economically advantageous.

16. Figure 2 is not clear and it is not necessary, since these results are already presented in Table 3. Please delete

Answer: The suggested figure has been deleted.

Round 2

Reviewer 2 Report

Comments and Suggestions for Authors

Necessary arrangements have been made. It is appropriate to publish.

Author Response

Dear Colleague,

We appreciate your detailed review of our article. Your insights have been extremely helpful in refining our manuscript. We have carefully considered your feedback and implemented the required revisions to improve the overall quality of our work.

Thank you for your thoughtful engagement with our research.

Warm regards,

Abbas Farshad

Reviewer 4 Report

Comments and Suggestions for Authors

Authors made the necessary amendments and I suggest the acceptance of their article. However, their strategy is not part of a biotechnology technique (L273-274)

Comments on the Quality of English Language

Minor editing of English language required

Author Response

Dear Colleague,

Thank you for taking the time to send us your thorough analysis of our article. Your input has proved invaluable in improving the content of our manuscript. We have carefully reviewed all of your suggestions and have made the required adjustments to elevate the quality of our work.

In response to the observation that the strategy is not classified as a biotechnological technique (L273-274), we have endeavored to reformulate the content and mark it in yellow within the manuscript.

Thank you for your insightful participation in our study.

Warm regards,

Abbas Farshad
